# Transformation of Amides to Thioamides Using an Efficient and Novel Thiating Reagent

**DOI:** 10.3390/molecules27238275

**Published:** 2022-11-27

**Authors:** Mohamed S. Gomaa, Gaber El Enany, Walid Fathalla, Ibrahim A. I. Ali, Samir. M. El Rayes

**Affiliations:** 1Department of Pharmaceutical Chemistry, College of Clinical Pharmacy, Imam Abdulrahman Bin Faisal University, P.O. Box 1982, Dammam 31441, Saudi Arabia; 2Department of Physics, College of Science and Arts in Uglat Asugour, Qassim University, Buraidah 52571, Saudi Arabia; 3Scientific Department, Faculty of Engineering, Port Said University, Port Said 42526, Egypt; 4Department of Chemistry, Faculty of Science, Suez Canal University, Ismailia 41522, Egypt

**Keywords:** *N*-aryl-substituted benzamides, benzothioamides, thiating reagent, *N*-isopropyldithiocarbamate isopropyl ammonium salt

## Abstract

A convenient protocol was developed for the transformation of *N*-aryl-substituted benzamides to *N*-aryl-substituted benzothioamides using *N*-isopropyldithiocarbamate isopropyl ammonium salt as a novel thiating reagent. The major advantages of this protocol are its *one-pot* procedure, short reaction times, mild conditions, simple work-up, high yields and pure products.

## 1. Introduction

Amides and their isosteric thioamides are very important biologically active compounds [1]. Benzothioamides are used as building blocks for the preparation of various compounds, especially those containing sulfur [2,3,4]. Many methods have been reported for the synthesis of substituted thiobenzamides from various substrates and reagents. Typically, Beckmann rearrangement is applied to prepare thioamides using ketoxime substrates in the presence of PSCl_3_ [5]_,_ in situ generated Appel’s salt, Mitsunobu’s zwitterionic adduct [6] or *O*,*O*-diethyl dithiophosphoric acid [7] as the dehydrating agents. Another interesting method for the introduction of thioamides is the modified Willgerodt–Kindler reaction using a three-component reaction: aniline, aldehydes, and elemental sulfur powder in the presence of sulfated tungstate catalyst as an acidic catalyst [8], Na_2_S as a basic catalyst [9] or sulfonic-acid-functionalized nano γ-Al_2_O_3_ [10], or in the absence of a catalyst [11,12]. Another method was carried out using thioacyl dithiophosphates for the thioacylation of amines to afford thioamides and water-soluble ammonium monothiophosphates [13,14]. Other methods involve *N*-substituted amide–thioamide transformation via thionation reagents such as Lawesson’s reagent (2,4-bis(4-ethoxyphenyl)-1,3-dithia-2,4-diphosphetane 2,4-disulfide) [15] and Berzelius reagent [16] (P_4_S_10_) in dry toluene, xylene or pyridine under reflux conditions. Some of these protocols are very interesting, showing high yields, but still have major drawbacks such as harsh reaction conditions, prolonged reaction times, expensive specific reagents, ultra-dry solvents and bad smell, which have limited their applications. A new protocol for thioamide preparation is highly needed. Our group developed a heterocyclic benzamide–thioamide transformation by chlorination followed by reaction with *N*-cyclohexyl dithiocarbamate cyclohexyl ammonium salt in chloroform for 12 h at 61 °C to afford heterocyclic thioamides in excellent yields [17,18,19,20]. As a part of our continuous efforts towards the improvement of synthetic methods, we now report a highly efficient protocol for open-chain amide–thioamide transformation via two-step, *one-pot* sequential reactions: first, chlorination of the benzanilides; second, reaction with an efficient and novel thiating reagent, *N*-isopropyldithiocarbamate isopropyl ammonium salt, in acetonitrile for 1 h. This reagent is prepared from simple commercial substrates, isopropyl amine and carbon disulfide.

## 2. Discussion

*N*-Cyclohexyl dithiocarbamate cyclohexyl ammonium salt **2** proved to be an interesting thiating reagent used in the preparation of a number of heterocyclic thioamides [17,18,19,20]. The thiating reagent *N*-cyclohexyl dithiocarbamate cyclohexyl ammonium salt **2** was simply prepared via a cyclohexyl amine reaction with carbon disulfide in water at room temperature for 2 h, Figure 1 [17]. 

Herein, we wish to examine the application of this thiating reagent to involve open-chain amide–thioamide transformation. Thus, the reaction of benzoyl chloride **3** with *p*-toluidine **4** in benzene in the presence of triethyl amine for 6 h afforded the model precursor *N*-(*p*-tolyl)benzamide (**5**). The reaction of *N*-(*p*-tolyl)benzamide (**5**) with thionyl chloride at 70 °C for 8 h gave *N*-(*p*-tolyl)benzimidoyl chloride (**6**), and thionyl chloride was then evaporated under reduced pressure. The imidoyl chloride **6** was kept under reduced pressure (6.7 KPa) at 120 °C for 2 h and was used in situ without further purification [21]. Finally, the in situ generated acetonitrile solution of imidoyl chloride **6** was treated under reflux conditions with **2** to afford a mixture of the desired compound, *N*-(*p*-tolyl)benzothioamide (**7**), and cyclohexylcarbamothioic-*N*-(*p*-tolyl)benzimidic thioanhydride (**8**), Figure 2. 

This behavior is most probably due to steric hindrance of the cyclohexyl moiety, which hinders the cyclohexyl amine proton abstraction by the imine nitrogen atom. Consequently, out of necessity, we should modify the structure of the thiating reagent. Recently, we reported an interesting method for the preparation of 2-arylquinazolin-4-amines by the reaction of *N*-(2-cyanophenyl)-substituted benzimidoyl isothiocyanates with isopropyl amine through 1-isopropyl-3-(2-(4-substitutedphenyl)quinazolin-4-yl)thiourea [22]. In accordance with these facts, we designed *N*-isopropyldithiocarbamate isopropyl ammonium salt **10** as our novel thiating reagent to solve the mentioned discrete problem. The reaction of three molar equivalents of isopropylamine **9** with one molar equivalent of carbon disulfide in ethyl acetate for 2 h precipitated pure *N*-isopropyldithiocarbamate isopropyl ammonium salt **10** in high yield, Figure 3. Compound **10** is hygroscopic and should be tightly packed.

Indeed, our hypothesis related to applying *N*-isopropyldithiocarbamate isopropyl ammonium salt **10** as a novel thiating reagent gave remarkable results. Thus, *N*-aryl-substituted benzamides **11**–**13**(**a**–**e**) were prepared by the reaction of carboxylic acid chlorides with anilines as described earlier. *N*-Aryl-substituted benzamides **11**–**13**(**a**–**e**) were chlorinated with thionyl chloride at 70 °C for 8 h, and the acetonitrile solutions of the in situ generated benzimidoyl chloride **14**–**16**(**a**–**e**) were treated with *N*-isopropyldithiocarbamate isopropyl ammonium salt (**10**) at room temperature for 1 h (TLC monitored). The reaction mixture was evaporated, and ethanol was added to give bright yellow crystals as the only isolated products, identified as *N*-aryl-substituted benzothioamides **17**–**19**(**a**–**e**), Figure 4. 

A total number of 15 *N*-aryl-substituted benzothioamides **17**–**19**(**a**–**e**) were obtained from *N*-aryl-substituted benzamides **11**–**13**(**a**–**e**) in a *one-pot* strategy and in excellent yields. This synthetic procedure using *N*-isopropyldithiocarbamate isopropyl ammonium salt **10** as the thiating reagent, besides solving all previously mentioned problems, has the advantage of applying the *one-pot* strategy for the amide–thioamide transformations, in addition to its operational simplicity, availability of substrates, reaction at room temperature and high yields in a short reaction time. 

A rational mechanism for this interesting protocol is given in Figure 5. The reaction of *N*-(*p*-tolyl)benzamide (**11b**) with thionyl chloride principally afforded *N*-(*p*-tolyl)benzimidoyl chloride (**14b**). The in situ generated **14b** solution in acetonitrile reacted with *N*-isopropyldithiocarbamate isopropyl ammonium salt (**10**) to principally afford *(E)-N*-(*p*-tolyl)benzimidic thioanhydride **I** as the most stable geometric isomer. Earlier results obtained by our group concerning similar structures showed that the reaction of *N*-(2-chloro-5-nitrophenyl)benzimidoyl isothiocyanate with *tert*-butyl amine principally afforded the intermediate (*E*)-*N*-(*tert*-butyl)carbamimidic-*N*-(2-chloro-5-nitrophenyl)benzimidic thioanhydride, which subsequently converted to the stable bis-[*N*-(2-chloro-5-nitrophenyl)benzimidoyl] sulfide as a Z-geometric isomer (the structure assignment of this compound was corroborated by X-ray crystallographic analysis) [23]. The tolyl residue directly attached to the imine nitrogen pushes electrons to it, which enhances the hydrogen bond acceptor character for this imine nitrogen atom. This behavior causes the nitrogen atom of the imine group to be able to abstract the isopropyl amine NH proton, and the residual electrons attack the thiocarbonyl group with consequent cleavage of the S–C bond, finally giving our desired product **17b** and isopropyl isothiocyanate, Figure 5. A similar result was obtained by our group for the rearrangement of 2-phenylquinazolin-4-yl cyclohexylcarbamodithioate to finally produce 2-phenylquinazoline-4(3*H*)-thione in the presence or absence of a base [17]. Also, a similar explanation was given by our group for the 2-arylquinazolin-4-amines from 1-isopropyl-3-(2-(4-substitutedphenyl)-quinazolin-4-yl)thiourea in the presence of isopropyl amine [22]. 

## 3. Conclusions

In this work, we successfully developed a facile and convenient general method for the conversion of *N*-aryl-substituted benzamides to *N*-aryl-substituted benzothioamides. This protocol consists of two steps in a *one-pot* strategy: first, we transformed *N*-aryl-substituted benzamides to benzimidoyl chloride derivatives by a reaction with thionyl chloride; second, the benzimidoyl chloride derivatives reacted with *N*-isopropyldithiocarbamate isopropyl ammonium salt as a novel thiating reagent to finally produce *N*-aryl-substituted benzothioamide. This method has the advantage of applying a *one-pot* strategy for amide–thioamide transformations, in addition to its operational simplicity, availability of substrates, reaction at room temperature and high yields in a short reaction time. 

## 4. Experimental

**General procedures.** Solvents were purified and dried by standard procedures. The boiling range of the petroleum ether used was 40–60 °C. Thin-layer chromatography (TLC): silica gel 60 F_254_ plastic plates (E. Merck, layer thickness 0.2 mm) detected by UV absorption. Elemental analyses were performed on a *Flash EA-1112* instrument at the Microanalytical laboratory, Faculty of Science, Suez Canal University, Ismailia, Egypt. Melting points were determined on a Buchi 510 melting-point apparatus, and the values are uncorrected. ^1^H and ^13^C NMR spectra were recorded at 400 MHz and 100 MHz, respectively (Bruker AC 400), in CDCl_3_ and DMSO solution with tetramethylsilane as an internal standard. The NMR analyses were performed at Faculty of Science, Sohag University. The thiating reagent *N*-cyclohexyl dithiocarbamate cyclohexyl ammonium salt (2) was obtained from cyclohexyl amine and carbon disulfide as described [17].


**Preparation of cyclohexylcarbamothioic-*N*-(*p*-tolyl)benzimidic thioanhydride (8).**


A mixture of *N*-(*p*-tolyl)benzoamide (**5**) (2.5 mmol) and thionyl chloride (5 mL) was heated at 70 °C for 8 h. The thionyl chloride was removed under reduced pressure and was heated at 120 °C under reduced pressure (50 mmHg) for an additional 2 h to give a clear yellowish-colored oil of benzimidoyl chloride **6**, which was not further purified and was used directly in the next step. To a solution of benzimidoyl chloride **6** (2.5 mmol) in acetonitrile (10 mL) was added 0.69 g (2.5 mmol) of *N*-cyclohexyl dithiocarbamate cyclohexyl ammonium salt (**2**). The reaction mixture was stirred at room temperature for 1 h. (TLC monitored), then heated for 2 h. The reaction mixture was evaporated under reduced pressure, and 25 mL of ethanol was added to the solid residue. The yellowish precipitate was filtered to give **8**. The crude compound was purified by crystallization from ethyl alcohol 95%. 

Yield 84% yellow crystals, mp 138–139 °C. ^1^H NMR spectrum, (400 MHz, CDCl_3_), δ, ppm (*J*, Hz): 7.89 (2H, d, *J* = 8.0, ArH); 7.82 (1H, bs, NH); 7.55–7.48 (5H, m, ArH); 7.19 (2H, d, *J* = 8.0, ArH); 3.87–3.67 (1H, m, CH); 2.36 (3H, s, CH_3_); 2.06–1.59 (4H, m, 2CH_2_); 1.44–1.20 (6H, m, 3CH_2_). Found, %: C, 68.36; H, 6.48; N, 7.46. For C_21_H_24_N_2_S_2_ (368.6). Calculated, %: C, 68.44; H, 6.56; N, 7.60; S, 17.40.


**General method for the preparation of thiating reagent *N*-isopropyldithiocarbamate isopropyl ammonium salt (10)**
**.**


To a mixture of isopropyl amine (60 mmol) and ethyl acetate (50 mL) was added carbon disulfide (21 mmol) dropwise. The reaction mixture was stirred at room temperature for 2 h. The white solid obtained was filtered, washed with ethyl acetate, dried and packed tightly, and it was pure enough for further reactions.

Yield 95% (acetonitrile) white crystals, mp 85–86 °C. ^1^H NMR spectrum, (400 MHz, CDCl_3_), δ, ppm (*J*, Hz): 8.19 (1H, bs, NH); 6.06–5.75 (3H, bs, 3NH); 3.88–3.75 (1H, m, CH); 3.72–3.51 (1H, m, CH); 1.43–1.08 (12H, m, 4CH_3_). ^13^C NMR spectrum, (100.0 MHz, CDCl_3_), δ, ppm: 211.8 (C=S); 52.7 (CH); 48.3 (CH); 24.5 (2CH_3_); 22.3 (2CH_3_). Found, %: C, 43.17; H, 9.23; N, 14.38. For C_7_H_18_N_2_S_2_ (194.4). Calculated, %: C, 43.26; H, 9.34; N, 14.41; S, 32.99.


**General method for the preparation of**
**
*N*
**
**-aryl substituted benzthioamide 17-19(a-e)**
**.**


A mixture of *N*-aryl-substituted benzamide **11–13**(**a–e**) (2.5 mmol) and thionyl chloride (5 mL) was heated at 70 °C for 8 h. The thionyl chloride was removed under reduced pressure and was heated at 120 °C under reduced pressure (50 mmHg) for an additional 2 h to give a clear yellowish-colored oil of benzimidoyl chloride **14–16**(**a–e**), which was not further purified and was used directly in the next step. To a solution of benzimidoyl chloride **14–16**(**a–e**) (2.5 mmol) in acetonitrile (10 mL) was added 0.49 g (2.5 mmol) of *N*-isopropyldithiocarbamate isopropyl ammonium salt (**10**). The reaction mixture was stirred at room temperature for 1 h (TLC monitored). The reaction mixture was evaporated under reduced pressure, and 25 mL of ethanol was added to the solid residue. The yellowish precipitate was filtered to give the desired *N*-aryl-substituted benzothioamides **17–19**(**a–e**). The crude compounds were pure enough for analytical purposes. Purification of the products for analysis was achieved by crystallization from the appropriate solvent. 


***N*-Phenylbenzothioamide (17a).**


Yield 89% (ethanol 95%) yellow crystals, mp 100–101 °C (lit. [24] 102 °C). ^1^H NMR spectrum, (400 MHz, CDCl_3_), δ, ppm (*J*, Hz): 9.06 (1H, bs, NH); 8.20-7.80 (4H, m, ArH); 7.64–7.29 (6H, m, ArH).^13^C NMR spectrum, (100.0 MHz, CDCl_3_), δ, ppm: 197.5 (C=S), 143.5, 139.6, 130.2, 129.3, 128.9, 127.6, 125.8, 124.7. Found, %: C, 73.15; H, 5.17; N, 6.53. For C_13_H_11_NS (213.3). Calculated, %: C, 73.20; H, 5.20; N, 6.57; S, 15.03.


***N*-(*p*-Tolyl)benzothioamide (17b).**


Yield 93% (ethanol 95%) yellow crystals, mp 126–127 °C (lit. [8] 130 °C). ^1^H NMR spectrum, (400 MHz, CDCl_3_), δ, ppm (*J*, Hz): 8.99 (1H, bs, NH); 7.88–7.77 (2H, m, ArH); 7.66 (2H, d, *J* = 8.0, ArH); 7.51 (2H, d, *J* = 8.0, ArH); 7.42–7.39 (3H, m, ArH); 2.37 (3H, s, CH_3_).^13^C NMR spectrum, (100.0 MHz, CDCl_3_), δ, ppm: 198.8 (C=S), 143.7, 136.7, 135.1, 130.6, 129.9, 128.4, 127.6, 125.9, 124.3, 21.7 (CH_3_). Found, %: C, 73.93; H, 5.74; N, 6.09. For C_14_H_13_NS (227.3). Calculated, %: C, 73.97; H, 5.76; N, 6.16; S, 14.10.


***N*-(*m*-Tolyl)benzothioamide (17c).**


Yield 76% (ethanol 95%) yellow crystals, mp 87–88 °C (lit. [25] 81 °C). ^1^H NMR spectrum, (400 MHz, CDCl_3_), δ, ppm (*J*, Hz): 9.00 (1H, bs, NH); 7.88–7.72 (2H, m, ArH); 7.67–7.54 (2H, m, ArH); 7.46–7.36 (3H, m, ArH); 7.31–7.18 (2H, m, ArH); 2.42 (3H, s, CH_3_). Found, %: C, 73.89; H, 5.72; N, 6.11. For C_14_H_13_NS (227.3). Calculated, %: C, 73.97; H, 5.76; N, 6.16; S, 14.10.


***N*-(*o*-Tolyl)benzothioamide (17d).**


Yield 84% (ethanol 95%) yellow crystals, mp 148–150 °C (lit. [24] 151 °C). ^1^H NMR spectrum, (400 MHz, CDCl_3_), δ, ppm (*J*, Hz): 7.86 (1H, d, *J* = 6.0, ArH); 7.81 (2H, d, *J* = 6.0, ArH); 7.59 (1H, bs, NH); 7.59–7.41 (3H, m, ArH); 7.22–7.15 (2H, m, ArH); 7.13 (1H, t, *J* = 6.0, ArH) 2.27 (3H, s, CH_3_).^13^C NMR spectrum, (100.0 MHz, CDCl_3_), δ, ppm: 198.2 (C=S), 142.9, 139.1, 133.6, 131.8, 131.3, 128.9,126.0, 125.6,125.1, 18.4 (CH_3_). Found, %: C, 73.85; H, 5.70; N, 6.12. For C_14_H_13_NS (227.3). Calculated, %: C, 73.97; H, 5.76; N, 6.16; S, 14.10.


***N*-(4-Methoxyphenyl)benzothioamide (17e).**


Yield 96% (ethanol 95%) yellow crystals, mp 130–131 °C (lit. [24] 12 7 °C). ^1^H NMR spectrum, (400 MHz, CDCl_3_), δ, ppm (*J*, Hz): 8.89 (1H, bs, NH); 7.87–7.76 (2H, m, ArH); 7.54 (2H, d, *J* = 8.0, ArH); 7.53–7.42 (3H, m, ArH); 6.88 (2H, d, *J* = 8.0, ArH); 3.73 (3H, s, OCH_3_).^13^C NMR spectrum, (100.0 MHz, CDCl_3_), δ, ppm: 199.6 (C=S), 158.6, 142.8, 131.4, 130.2, 128.3, 127.5, 125.3, 113.8, 55.3 (OCH_3_). Found, %: C, 69.03; H, 5.36; N, 5.72. For C_14_H_13_NOS (243.3). Calculated, %: C, 69.11; H, 5.39; N, 5.76; S, 13.18.


**4-Methoxy-*N*-phenylbenzothioamide (18a).**


Yield 82% (ethanol 95%) yellow crystals, mp 145–146 °C (lit. [8] 150 °C). ^1^H NMR spectrum, (400 MHz, CDCl_3_), δ, ppm (*J*, Hz): 7.86 (2H, d, *J* = 6.0, ArH); 7.71 (1H, bs, NH); 7.45–7.36 (2H, m, ArH); 7.34–7.29 (3H, m, ArH); 6.93 (2H, d, *J* = 6.0, ArH); 3.82 (3H, s, OCH_3_).^13^C NMR spectrum, (100.0 MHz, CDCl_3_), δ, ppm: 199.4 (C=S), 158.4, 139.1, 134.5, 129.0, 127.2, 126.9, 123.4, 114.1, 55.5 (OCH_3_). Found, %: C, 69.09; H, 5.38; N, 5.72. For C_14_H_13_NOS (243.3). Calculated, %: C, 69.11; H, 5.39; N, 5.76; S, 13.18.


**4-Methoxy-*N*-(*p*-tolyl)benzothioamide (18b).**


Yield 93% (ethanol 95%) yellow crystals, mp 172–173 °C (lit. [26] 174 °C). ^1^H NMR spectrum, (400 MHz, CDCl_3_), δ, ppm (*J*, Hz): 7.85 (2H, d, *J* = 6.0, ArH); 7.73 (1H, bs, NH); 7.52 (2H, d, *J* = 6.0, ArH); 7.19 (2H, d, *J* = 6.0, ArH); 6.99 (2H, d, *J* = 6.0, ArH); 3.73 (3H, s, OCH_3_); 2.36 (3H, s, CH_3_).^13^C NMR spectrum, (100.0 MHz, CDCl_3_), δ, ppm: 200.5 (C=S), 158.7, 136.6, 135.8, 134.1, 129.9, 127.4 124.7, 113.8, 55.7 (OCH_3_), 21.5 (CH_3_). Found, %: C, 69.94; H, 5.76; N, 5.38. For C_15_H_15_NOS (257.4). Calculated, %: C, 70.01; H, 5.88; N, 5.44; S, 12.46.


**4-Methoxy *N*-(*m*-tolyl)benzothioamide (18c).**


Yield 86% (ethanol 95%) yellow crystals, mp 132–133 °C. ^1^H NMR spectrum, (400 MHz, CDCl_3_), δ, ppm (*J*, Hz): 7.82 (2H, d, *J* = 6.0, ArH); 7.67 (1H, bs, NH); 7.62-7.54 (2H, m, ArH); 7.31-7.11 (2H, m, ArH); 6.93 (2H, d, *J* = 6.0, ArH); 3.86 (3H, s, OCH_3_); 2.39 (3H, s, CH_3_).^13^C NMR spectrum, (100.0 MHz, CDCl_3_), δ, ppm: 200.2 (C=S), 158.5, 138.9, 136.3, 135.4, 134.4, 129.8, 127.6, 125.9, 124.5, 114.6, 21.2 (CH_3_). Found, %: C, 69.87; H, 5.74; N, 5.36. For C_15_H_15_NOS (257.4). Calculated, %: C, 70.01; H, 5.88; N, 5.44; S, 12.46.


**4-Methoxy-*N*-(*o*-tolyl)benzothioamide (18d).**


Yield 94% (ethanol 95%) yellow crystals, mp 125–126 °C (lit. [27] 119 °C). ^1^H NMR spectrum, (400 MHz, CDCl_3_), δ, ppm (*J*, Hz): 7.94 (1H, d, *J* = 6.0, ArH); 7.88 (2H, d, *J* = 6.0, ArH); 7.64 (1H, bs, NH); 7.29–7.24 (2H, m, ArH); 7.13 (1H, t, *J* = 6.0, ArH); 7.00 (2H, d, *J* = 6.0, ArH); 3.98 (3H, s, OCH_3_); 2.36 (3H, s, CH_3_).^13^C NMR spectrum, (100.0 MHz, CDCl_3_), δ, ppm: 199.1 (C=S), 159.3, 139.2, 136.5, 134.3, 133.6, 130.1, 127.4, 126.4, 125.6, 113.9, 55.5 (OCH_3_), 18.9 (CH_3_). Found, %: C, 69.92; H, 5.79; N, 5.41. For C_15_H_15_NOS (257.4). Calculated, %: C, 70.01; H, 5.88; N, 5.44; S, 12.46.


**4-Methoxy-*N*-(4-methoxyphenyl)benzothioamide (18e).**


Yield 90% (ethanol 95%) yellow crystals, mp 149–150 °C (lit. [8] 152 °C). ^1^H NMR spectrum, (400 MHz, CDCl_3_), δ, ppm (*J*, Hz): 7.75 (2H, d, *J* = 6.0, ArH); 7.61 (1H, bs, NH); 7.44 (2H, d, *J* = 6.0, ArH); 6.89 (2H, d, *J* = 6.0, ArH); 6.82 (2H, d, *J* = 6.0, ArH); 3.82 (3H, s, OCH_3_); 3.74 (3H, s, OCH_3_). ^13^C NMR spectrum, (100.0 MHz, CDCl_3_), δ, ppm: 202.1 (C=S), 159.7, 157.6, 134.1, 131.3 127.8, 125.6, 114.4, 113.7, 55.6 (OCH_3_), 55.2 (OCH_3_). Found, %: C, 65.88; H, 5.51; N, 5.04. For C_15_H_15_NO_2_S (273.4). Calculated, %: C, 65.91; H, 5.53; N, 5.12; S, 11.73.


**4-Chloro-*N*-phenylbenzothioamide (19a).**


Yield 92% (ethanol 95%) yellow crystals, mp 157-158 °C (lit. [11] 153 °C). ^1^H NMR spectrum, (400 MHz, CDCl_3_), δ, ppm (*J*, Hz): 9.03 (1H, bs, NH); 7.67 (2H, d, *J* = 6.0, ArH); 7.45–7.36 (4H, m, ArH); 7.34-7.29 (3H, m, ArH).^13^C NMR spectrum, (100.0 MHz, CDCl_3_), δ, ppm: 198.6 (C=S), 140.3, 139.8, 135.3, 129.0, 128.5, 127.9, 127.3, 123.6. Found, %: C, 62.89; H, 4.05; N, 5.61. For C_13_H_10_ClNS (247.7). Calculated, %: C, 63.03; H, 4.07; Cl, 14.31; N, 5.65; S, 12.94.


**4-Chloro-*N*-(*p*-tolyl)benzothioamide (19b).**


Yield 96% (ethanol 95%) yellow crystals, mp 150–151 °C. ^1^H NMR spectrum, (400 MHz, CDCl_3_), δ, ppm (*J*, Hz): 9.03 (1H, bs, NH); 7.71 (2H, d, *J* = 6.0, ArH); 7.52 (2H, d, *J* = 6.0, ArH); 7.42 (2H, d, *J* = 6.0, ArH); 7.19 (2H, d, *J* = 6.0, ArH); 2.37 (3H, s, CH_3_).^13^C NMR spectrum, (100.0 MHz, CDCl_3_), δ, ppm: 199.5 (C=S), 141.2, 136.3, 135.6, 134.7, 129.2, 128.9, 127.6, 124.5, 21.3 (CH_3_). Found, %: C, 64.14; H, 4.59; N, 5.28. For C_14_H_12_ClNS (261.8). Calculated, %: C, 64.24; H, 4.62; Cl, 13.54; N, 5.35; S, 12.25.


**4-Chloro-*N*-(*m*-tolyl)benzothioamide (19c).**


Yield 98% (ethanol 95%) yellow crystals, mp 136-137 °C. ^1^H NMR spectrum, (400 MHz, CDCl_3_), δ, ppm (*J*, Hz): 9.01 (1H, bs, NH); 7.67–7.54 (4H, m, ArH); 7.39 (2H, d, *J* = 6.0, ArH); 5.31–7.13 (2H, m, ArH); 2.36 (3H, s, CH_3_).^13^C NMR spectrum, (100.0 MHz, CDCl_3_), δ, ppm: 199.6 (C=S), 141.5, 138.4, 136.7, 135.6, 134.9, 129.6, 129.0, 127.2, 125.8, 123.4. 21.5 (CH_3_). Found, %: C, 64.18; H, 4.54; N, 5.29. For C_14_H_12_ClNS (261.8). Calculated, %: C, 64.24; H, 4.62; Cl, 13.54; N, 5.35; S, 12.25.


**4-Chloro-*N*-(*o*-tolyl)benzothioamide (19d).**


Yield 73% (ethanol 95%) yellow crystals, mp 129-130 °C. ^1^H NMR spectrum, (400 MHz, CDCl_3_), δ, ppm (*J*, Hz): 9.03 (1H, bs, NH); 7.94 (1H, d, *J* = 6.0, ArH); 7.67 (2H, d, *J* = 6.0, ArH); 7.39 (2H, d, *J* = 6.0, ArH); 7.31–7.24 (2H, m, ArH); 7.16 (1H, t, *J* = 6.0, ArH); 2.36 (3H, s, CH_3_). ^13^C NMR spectrum, (100.0 MHz, CDCl_3_), δ, ppm: 199.3 (C=S), 140.5, 139.4, 136.5, 134.5, 133.8, 130.5, 128.8, 127.9, 127.3, 125.6, 18.5 (CH_3_). Found, %: C, 64.12; H, 4.48; N, 5.31. For C_14_H_12_ClNS (261.8). Calculated, %: C, 64.24; H, 4.62; Cl, 13.54; N, 5.35; S, 12.25.


**4-Chloro-*N*-(4-methoxyphenyl)benzothioamide (19e).**


Yield 85% (ethanol 95%) yellow crystals, mp 166–167 °C (lit. [28] 172 °C). ^1^H NMR spectrum, (400 MHz, CDCl_3_), δ, ppm (J, Hz): 8.89 (1H, bs, NH); 7.82 (2H, d, *J* = 6.0, ArH); 7.64 (2H, d, *J* = 6.0, ArH); 7.36 (2H, d, *J* = 6.0, ArH); 6.91 (2H, d, *J* = 6.0, ArH); 3.75 (3H, s, OCH_3_).^13^C NMR spectrum, (100.0 MHz, DMSO), δ, ppm: 199.8 (C=S), 158.9, 140.4, 134.2, 131.9, 128.6, 127.6, 126.3, 114.5, 55.4 (OCH_3_). Found, %: C, 60.49; H, 4.32; N, 4.87. For C_14_H_12_NClNOS (277.8). Calculated, %: C, 60.54; H, 4.35; Cl, 12.76; N, 5.04; S, 11.54.

## Data Availability

All data are already presented in detail in the manuscript in each section under each corresponding title.

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
