# Peer review of "Transformation of Amides to Thioamides Using an Efficient and Novel Thiating Reagent"

_molecules, 2022, doi:10.3390/molecules27238275_

Round 1

Reviewer 1 Report

Review of the Article

Transformation of amides to thioamides using efficient and
novel thiating reagent
 by Mohamed Sayed Gomaa, Gaber El Enany,
 Walid Fathalla Ibrahim A. I. Ali,d Samir. M. El Rayes*
Authors described the transformation of N-aryl substituted benzamides to N-aryl substituted benzothioamides using N-isopropyldithiocarbamate isopropyl ammonium salt 2.

In the work presented for review, the Authors' self-plagiarism can be noticed. Diagram showing the synthesis of N-cyclohexyl dithiocarbamate cyclohexyl ammonium salt number 2 as well as in the Experimental section for the general synthetic procedure has already been described and published twice, See references 17 and 19 in the References.
Is there any sense in describing the same for the third time?
In my opinion, it is enough to mention the above-mentioned items (17 and 19).
In addition, R1, R2, R3, R4 are incorrectly listed in Scheme 4.
How is it possible that for starting material number 11 - 13, R1 - R4 = H, or methyl, while in product number 17 - 19, also Cl and methoxy appear in the phenyl ring (which is not a reactive center)?
This should be clearly corrected.In scheme 5:Rational mechanism of N-(p-tolyl)benzamide thiation the rest from the ammonium salt 10 is omited.It means formation of the isopropyl ammonium chloride as a side product should be added.
There are also too many auto-citations in the work presented for review. These citations should be corrected to the necessary minimum, so that the work would not be artificially abundant.a side product should be added.
I suggest that you send it after proofreading to a specialized journal, e.g. Heteroatom;J. Heterocyclic Chemistry; Beilstein J.Org.Chem; Journal of Sulfur Chemistry or Phosphorus, Sulfur and Related Elements. The work should be published as a short communication.

Author Response

Reviewer 1

In the work presented for review, the Authors' self-plagiarism can be noticed. Diagram showing the synthesis of N-cyclohexyl dithiocarbamate cyclohexyl ammonium salt number 2 as well as in the Experimental section for the general synthetic procedure has already been described and published twice, See references 17 and 19 in the References.

Is there any sense in describing the same for the third time?

In my opinion, it is enough to mention the above-mentioned items (17 and 19(

Response

The description of synthesis was removed and we inserted the references numbers

In addition, R1, R2, R3, R4 are incorrectly listed in Scheme 4

How is it possible that for starting material number 11 - 13, R1 - R4 = H, or methyl, while in product number 17 - 19, also Cl and methoxy appear in the phenyl ring (which is not a reactive center(

Response

We have to explain the schemes in details for the reviewer to remove any confusion. The following draw explain the scheme in details  

This should be clearly corrected. In scheme 5:Rational mechanism of N-(p-tolyl)benzamide thiation the rest from the ammonium salt 10 is omitted. It means formation of the isopropyl ammonium chloride as a side product should be added

Response

Corrected

.

There are also too many auto-citations in the work presented for review. These citations should be corrected to the necessary minimum, so that the work would not be artificially abundant. A side product should be added

Response

We removed some references of self-citation

Reviewer 2 Report

The authors of this manuscript developed an approach to the synthesis of N-aryl-substituted  benzothioamides from the corresponding amides using a novel reagent N-isopropyldithiocarbamate isopropyl ammonium salt. The method is easy to carry out and it gives the desired products in high yields in short reactions. Because of the novelty, the manuscript is suggested for publication. However, the writing of the manuscript is unsatisfactory. This referee is under the impression that the authors were in a rush and put together the manuscript without the necessary care and attention. Considering the following remarks, significant improvements are needed before publication can be recommended.

1. Remarks

i) Introduction (corrected)

sulfonic acid/ Another method not Other method/ in dry toluene/ drawbacks such as harsh reaction conditions/ is highly needed/ we now report a highly efficient/ using two-step, one-pot sequential reactions; ask the help of a professional!!!

ii) p.2, line 1: for 1h should read 1 h; Scheme 1: 25oC should read 25 oC./ using two steps one-pot sequential reactions/ in situ not insitu/p.2: which tangle… which hinders… would be better/ Aryl substituted should read Aryl-substituted

Make correction throughout the manuscript!

iii) English usage – a few examples:

p.2, above Scheme 2: was refluxed… The verb „to reflux” does not exist in English! Use …was treated under reflux conditions

p.3, below Scheme 4: This synthetic procedure…has the advantage

Conclusions: N-aryl-substituted benzamides to N-aryl-substituted benzothioamides (plural)

4.1: Solvents were purified (plural)

iv) References

The authors use abbreviated given names except in refs 17-20 and 24,25? Furthermore, compare author names in ref. 1–16 to ref. 12 etc.

Author Response

Reviewer 2

The authors of this manuscript developed an approach to the synthesis of N-aryl-substituted  benzothioamides from the corresponding amides using a novel reagent N-isopropyldithiocarbamate isopropyl ammonium salt. The method is easy to carry out and it gives the desired products in high yields in short reactions. Because of the novelty, the manuscript is suggested for publication. However, the writing of the manuscript is unsatisfactory. This referee is under the impression that the authors were in a rush and put together the manuscript without the necessary care and attention. Considering the following remarks, significant improvements are needed before publication can be recommended.

  1. 1. Remarks

  1. i) Introduction (corrected)

sulfonic acid/ Another method not Other method/ in dry toluene/ drawbacks such as harsh reaction conditions/ is highly needed/ we now report a highly efficient/ using two-step, one-pot sequential reactions; ask the help of a professional!!!

Response

Done

  1. ii) p.2, line 1: for 1h should read 1 h; Scheme 1: 25oC should read 25 oC./ using two steps one-pot sequential reactions/ in situ not insitu/p.2: which tangle… which hinders… would be better/ Aryl substituted should read Aryl-substituted

Make correction throughout the manuscript!

Response

Done

iii) English usage – a few examples:

p.2, above Scheme 2: was refluxed… The verb „to reflux” does not exist in English! Use …was treated under reflux conditions

Response

Done

p.3, below Scheme 4: This synthetic procedure…has the advantage

Response

Done

Conclusions: N-aryl-substituted benzamides to N-aryl-substituted benzothioamides (plural)

Solvents were purified (plural1:4))

Response

Done

  1. iv) References

The authors use abbreviated given names except in refs 17-20 and 24,25? Furthermore, compare author names in ref. 1–16 to ref. 12 etc

Response

Done

Round 2

Reviewer 1 Report

                                      Review  of the paper

Transformation of amides to thioamides using efficient and novel thiating                                                  reagent.

Unfortunatelly I can not find  attached the cover letter in which Authors addres the references comments.

In scheme 4 under the reaction pathway,in the table R-1, R-2, R-3 and R-4, are still not clearly described.  

Author Response

Reviewer 1

In addition, R1, R2, R3, R4 are incorrectly listed in Scheme 4

How is it possible that for starting material number 11 - 13, R1 - R4 = H, or methyl, while in product number 17 - 19, also Cl and methoxy appear in the phenyl ring (which is not a reactive center(

Response

"we attached a complete file with a scheme we hope this attached file arrive this time" 

Reviewer 2 Report

The authors have addressed all remarks of my evaluation and made satisfactory changes accordingly. Now publication is suggested without any further reservation.

Author Response

we deeply thank the full respectafull reviewers for their valuabale notes